# Determinants of COVID-19 vaccine acceptance and hesitancy among adolescents and youths aged 10-35 years in sub-Saharan African countries: A systematic review and meta-analysis

Allan Mayaba Mwiinde[1,2]*, Patrick Kaonga[1], Choolwe Jacobs[1], Joseph Mumba Zulu[3,4], Isaac Fwemba[5]

1 Department of Epidemiology and Biostatistics, School of Public Health, University of Zambia, Lusaka, Zambia, 2 Department of Public Health, Mazabuka Municipal Council, Mazabuka, Zambia, 3 Department of Health Promotion, School of Public Health, University of Zambia, Lusaka, Zambia, 4 Department of Health Policy and Management, School of Public Health, University of Zambia, Lusaka, Zambia, 5 Department of Medical Education, School of Medicine, University of Zambia, Lusaka, Zambia

* mayabamwiinde@gmail.com

## Abstract

The COVID-19 pandemic has overwhelmed health systems, especially in sub-Saharan African countries. Vaccination is one of the easily accessible interventions that can help reduce the burden on the health system. However, vaccination coverage remains low in sub-Saharan African countries. The determinants of vaccine acceptance and hesitancy among adolescents and youths remain unknown. Therefore, this study explored the pooled prevalence and determinants of COVID-19 vaccine acceptance and hesitancy among adolescents and youths in sub-Saharan African Countries. A systematic literature search of Scopus, PubMed Central, PubMed, Embase, African Journal Online, Research 4 Life, Embase, and Google Scholar was performed from 6th May to 31st December 2023, using developed keywords with a focus on sub-Saharan African countries. Twenty-three (N = 23) studies were finally selected for analysis. The pooled prevalence of vaccine acceptance among adolescents and youths was 38.7% (n = 23). The subgroup analysis of the pooled prevalence of acceptance among adolescents was 36.1% (n = 36.1) while youths were 42% (n = 10). At the region level, West Africa had 42.2% (n = 13), East Africa had 39.8% (n = 6), Central Africa had 33% (n = 1), and Southern Africa had 24.2% (n = 3). The determinants of vaccine acceptance were the desire for self-immunity (AOR = 1.97, 95%, CI, 1.083.47, $I^2$ = 94.15%, p < 0.05), receiving Health Officers' information (AOR = 4.36, 95%, CI, 2.28-8.32, $I^2$ = 97.74, p < 0.001), the effectiveness of COVID-19 vaccine (AOR = 2.14, 95%, CI, 1.14-4.05, $I^2$ = 97.4%, p < 0.05). The odds of having an unconfirmed source of information (AOR = 0.22, 95% CI, 0.10-0.45, $I^2$ = 94.09%, p< 0.001) was responsible for vaccine hesitancy. The findings indicate the low pooled prevalence of COVID-19 vaccine acceptance and high levels of hesitancy among adolescents and youths in sub-Saharan African countries. Therefore, there is a need to ensure that extensive research is undertaken into age-appropriate health

**Data Availability Statement:** All relevant data are within the paper and its Supporting Information files.

**Funding:** This work was funded by a grant from NORHED II portfolio 2021-2026 awarded to AMM (70324). The funders had no role in study design, data collection and analysis, decision to publish, or preparation of the manuscript.

**Competing interests:** The authors have declared that no competing interests exist.

**Abbreviations:** COVID-19, Corona Virus 2019; PRISMA, Preferred Reporting Items for Systematic Reviews and Meta-Analyses; SAGE, Strategic Advisory Group of Experts on Immunization; SARS-COV-2, Severe Acute Respiratory Syndrome Coronavirus 2; WHO, World Health Organization.

promotion messages and strategies to encourage the uptake of vaccines. PROSPERO ID number CRD42023403071.

## Introduction

Since the outbreak of the COVID-19 pandemic caused by the severe acute respiratory syndrome coronavirus 2 (SARS-COV-2) in Wuhan Hubei province of China an epidemic center in 2020, the World Health Organization declared it a global pandemic on the 11[th] March 2020 [1]. During the time COVID-19 was declared a global pandemic very little was known about the virus in terms of the dynamics of transmission, its incubation period, and the required medication as well as the type of vaccination, among others [2]. Since then, nations have devised various control strategies to curb the spread of COVID-19 in communities [3].

The newly developed COVID-19 vaccine has demonstrated effectiveness in reducing the severity and risk of COVID-19 occurrence globally among the human population [4]. The recorded mortalities from COVID-19 are high among adults of the age 65 years as such the focus of vaccination has been on this age group [5]. However, youths and adolescents still transmit the COVID-19 virus to the older population [6]. Complications, arising from COVID-19 infection have been recorded among adolescents and youths such as multisystem inflammatory syndrome and long COVID syndrome [7].

In sub-Saharan Africa, the first case of COVID-19 was reported on the 27th of February in Nigeria [8]. Sub-Saharan African countries have on average a young population [9]. Although vaccination is important in all age groups there have been few studies that have focused on adolescents and youths regarding vaccine hesitancy and acceptance [10]. Adolescence and youth are unique stages of human development and an important time for laying the foundations of good health [11]. Studies indicate that the continent is vulnerable to emerging infectious diseases such as COVID-19 since the population has not developed a full immune system against such infectious diseases [9]. The COVID-19 pandemic has overwhelmed health systems everywhere including sub-Saharan African countries [12]. Vaccination is one of the easily accessible and resilient interventions that can help reduce the burden on the health system [13]. However, the uptake of vaccines has been a challenge due to concerns such as safety and efficacy [14]. The success of vaccination relies on increasing vaccination coverage for the population to be susceptible to COVID-19 and to reduce the determinants of vaccine hesitancy [15]. If not addressed these determinants of hesitancy have the potential to pose global public health challenges and facilitate the emergence of new COVID-19 variants [16].

As of February 2024, the COVID-19 vaccination rate in Africa was at 51.85% while sub-Saharan African countries ranged from 0.29% in Burundi to more than 82% in Seychelles and Mauritius respectively [17]. This calls for extensive research to address the barriers to vaccine uptake and hesitancy in Africa. Some of the issues that shape vaccine hesitancy in sub-Saharan African countries include inadequate knowledge of the efficacy of vaccines which are the most effective public health interventions and have significantly helped control the morbidity, and mortality of communicable diseases [18]. Sub-Saharan African countries are also associated with historical, and cultural dynamics that are obstacles to increasing vaccine hesitancy among communities [18]. With the availability of information technology, global studies have established that determinants such as social media are major sources of information contributing to hesitancy while determinants of vaccine uptake include old age, education, demography, employment, etc, which differ from one place to another [19, 20].

Many review studies and meta-analyses conducted at the sub-Saharan African level have focused more on the determinants of vaccine acceptance and hesitancy at the population level, with a focus on adults however, very few studies have been conducted among adolescents and youths who make up around 70% of the sub-Saharan African population [21–23].

Vaccination of adolescents and youths should be considered a top priority targeting the primary vaccination and secondary vaccination booster vaccines in any epidemic in order to improve or archive herd immunity in the quickest possible time, as they carry the highest incidence of COVID-19 infection [24, 25]. Therefore, in this systematic review and meta-analysis, we aim to critically synthesize evidence on the pooled prevalence and determinants of COVID-19 vaccine acceptance and hesitancy among adolescents and youths in Sub-Saharan African countries. This is because adolescents and youths have a high risk of COVID-19 infection and the potential for high transmission rate in the communities due to prospects for social and economic advancement, causing continued circulation of the virus within the population [26].

## Methodology

### Search methods and eligibility

The study used the Preferred Reporting Items for Systematic Reviews and Meta-Analysis (PRISMA) guidelines to review the articles of the study [27]. The authors checked the PROSPERO database available at https://www.crd.york.ac.uk/prospero/ to verify whether there were any published articles or ongoing research projects similar to the topic to avoid duplication. The findings revealed that there were no articles or projects with a similar title in the database. The study was registered on PROSPERO ID number CRD42023403071.

The systematic review and meta-analysis of the studies conducted focused on sub-Saharan African countries assessing the pooled estimates and determinants of COVID-19 acceptance and hesitancy among adolescents and youths.

### Inclusion criteria

Due to the difference in the use of the official language, only articles written in English were considered. The date for the evaluation of the manuscript was limited to February 2020 when the first case of COVID-19 was reported in Africa until the last date of the search 31 December 2023. Only published articles with a focus on adolescents and youths (10-35 years) in a country belonging to the sub-Saharan African region were considered in the study. All published articles conducted using a cross-sectional study design are included in the review and meta-analysis.

### Exclusion criteria

Abstracts, articles, without full text, systematic reviews, conference papers, editorial letters, protocols, program evaluation report, qualitative studies, programme evaluation reports, and studies that were considered with low quality in the assessment were excluded from the study. Abstracts, studies without full texts, conference papers, editorials, letters, protocols, program evaluation reports, systematic reviews, trials, and qualitative studies were excluded from this study. In addition, studies with a high risk of bias or scored less than 50% on the critical appraisal checklist were excluded.

### Research questions

The research questions considered for this systematic review included the following. What is the prevalence of COVID-19 vaccine acceptance and hesitancy among adolescents and youths

in Sub-Saharan African countries? What is the extent of variation in the COVID-19 vaccine's acceptance and hesitancy rate among adolescents, youths and regions in sub-Saharan Africa countries? What are the determinants of COVID-19 vaccine acceptance and hesitancy among adolescents and youths in sub-Saharan African Countries?

## Outcome measurements

The study has three major outcomes. The primary outcomes were the prevalence of COVID-19 vaccine acceptance and hesitancy among adolescents and youths. Acceptance was defined as the proportion of participant's willingness or intention to receive a vaccine when it is available. Hesitancy was defined as the refusal or denial of uptake of the vaccine when it is readily available. All included studies asked their participants about their willingness to take the COVID-19 vaccine with a "yes" indicating acceptance and "no" indicating hesitancy". The third outcome was the determinants of COVID-19 vaccine acceptance and hesitancy among the participants in the studies.

Adolescence was defined as the phase of life between childhood and adulthood, from ages 10 to 19 [11]. In this research, the African Charter was used to define youth as people between the ages of 15-35 years [28].

## Information sources

A literature search was conducted using African, Journal Online, Embase, Google Schooler, PubMed Central, Research 4 Life, Scopus were used to search for the literature to be extracted. The study articles focused on all sub-Saharan African countries till the last date of the search 31[st] December, 2023 were included in the study. The searches were re-run prior to the final analysis and any further studies identified meeting the inclusion criteria were retrieved and included in the final manuscript.

## Search strategy

Literature was extracted by two independent authors M.M.A and I.F who were responsible for identifying relevant literature and extracting relevant data. Where there was disagreement, a third party (JZ) was consulted to determine the article's eligibility. The following search strategy was used: "COVID-19" OR "SARS-CoV-2" AND ''Vaccine" AND "Willingness" OR "Acceptance" OR "Intention" OR "Hesitancy" OR "Refusal" AND "Associated factors" OR "Determinants" OR "Predictors" AND "Adolescents" AND "Young Adults" AND "Youths" AND "University student" AND "African countries" OR "Sub-Saharan countries". The search string was developed using "AND" and "OR" Boolean operators. For PubMed searching, we have used this searching formulation: (((((((((((((COVID-19) OR (Corona Virus)) OR (SARS COV-2)) OR (Severe Acute Respiratory Syndrome)) AND (Vaccine)) OR (Immunization)) OR (Covid-19 Vaccine)) AND (Acceptance)) OR (Willingness)) OR (Intention)) OR (Uptake)) AND (Adolescents)) AND (Young Adults)) OR (Youths)) AND (Sub-Saharan Africa)) OR (African Countries).

## Study screening

Mendeley referencing manager was used to store all the references extracted which were able to transpose the data into the Microsoft Excel sheet which was used to remove duplication of the manuscript.

## Selection process

Two independent authors, A.M.M and I.F, performed the selection process for the systematic review and meta-analysis. They independently assessed the full texts of potential eligibility studies. Any differences or discrepancies between the two authors were discussed either online or in person. If the two authors could not reach an agreement, a third reviewer (J.Z) was invited to assess the articles and make the final determination of inclusion or exclusion of the manuscript. The titles of each manuscript were used to identify and extract relevant information. Once the title indicated that the abstract discussed COVID-19 vaccine uptake and hesitancy in any of the sub-Saharan African countries the full reference including the author, year, title, and abstract was obtained for further evaluation.

## Quality assessment

The quality of the analytical cross-sectional study was assessed using the Joanna Briggs Institute (JBI) Critical Appraisal Tool for Cross-Sectional Research was used in the study [29]. The authors assessed the quality of the manuscript which includes (i.e. inclusion and exclusion criteria, methodological quality, sample selection, sample size, reliability and validity, the confounding factors of the study used, and the correct statistical analysis). Eligibility and exclusion of studies were based on their titles and abstracts. The assessment checklist consisted of nine items, which evaluated aspects such as the appropriateness of the sample frame, selection methods, sample size, participant and setting characteristics statistical investigations, appropriate techniques valid measurements, appropriate statistical analysis; and an adequate response rate. Each item was assigned a response of a 'yes' "no" "unclear" or "not applicable." A "yes" response was marked in green, a "no" response in red, and the "unclear" response in grey (Table 1 and S1 Table) [30–44].

## Data collection process and risk of bias

Two independent researchers co-assessed the extracted data to ensure accuracy. The two authors then conducted a full-text assessment of potentially eligible studies independently. Any differences in opinions were resolved through discussion between the two researchers to

**Table 1. Critical appraisal tool for cross-section and research.**

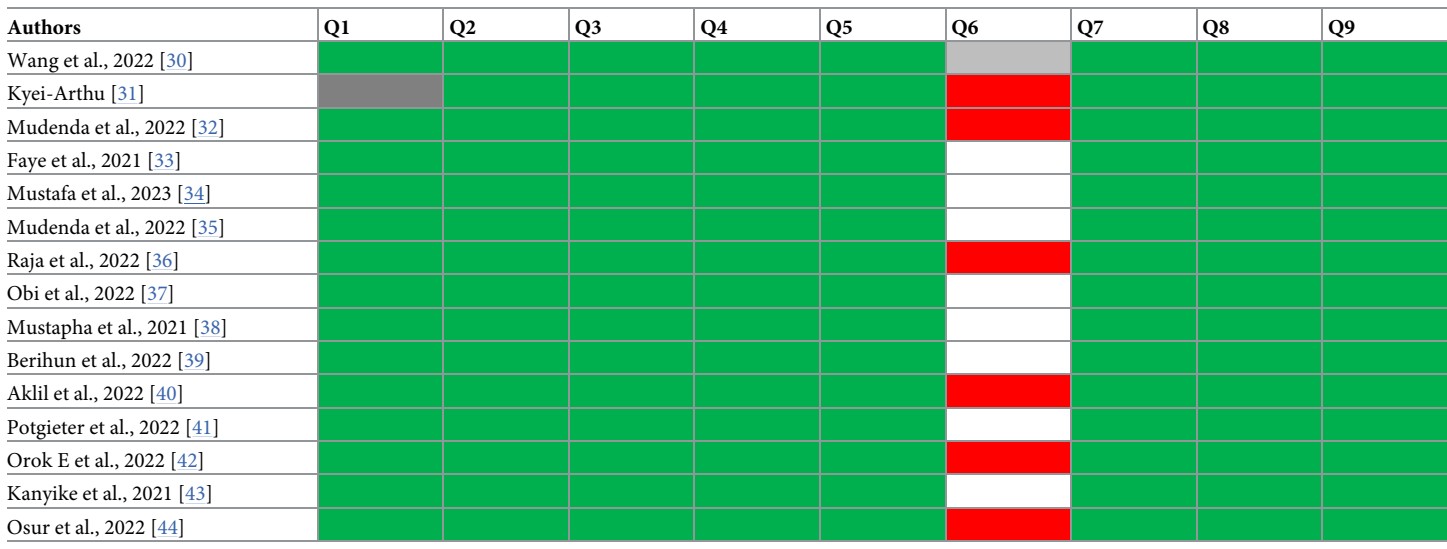

| Authors | Q1 | Q2 | Q3 | Q4 | Q5 | Q6 | Q7 | Q8 | Q9 |
|---|---|---|---|---|---|---|---|---|---|
| Wang et al., 2022 [30] | | | | | | | | | |
| Kyei-Arthu [31] | | | | | | | | | |
| Mudenda et al., 2022 [32] | | | | | | | | | |
| Faye et al., 2021 [33] | | | | | | | | | |
| Mustafa et al., 2023 [34] | | | | | | | | | |
| Mudenda et al., 2022 [35] | | | | | | | | | |
| Raja et al., 2022 [36] | | | | | | | | | |
| Obi et al., 2022 [37] | | | | | | | | | |
| Mustapha et al., 2021 [38] | | | | | | | | | |
| Berihun et al., 2022 [39] | | | | | | | | | |
| Aklil et al., 2022 [40] | | | | | | | | | |
| Potgieter et al., 2022 [41] | | | | | | | | | |
| Orok E et al., 2022 [42] | | | | | | | | | |
| Kanyike et al., 2021 [43] | | | | | | | | | |
| Osur et al., 2022 [44] | | | | | | | | | |

ensure that all the relevant data was extracted. Except where the two do not agree a third researcher was involved in making a final decision if needed. The extracted study characteristics included but were not limited to study design and sample size. Location, survey timescale, method of recruitment of the participants, and demographic characteristics. The prevalence of vaccine hesitancy and acceptance was also recorded.

## Measure of effect

The analysis was based on a total of twenty-three included studies. The event rate was used as the effect size index and all the raw variables were converted to event rates. A random-effects model was employed to determine the pooled prevalence of vaccine acceptance and hesitancy. Studies in the analysis represented a random sample from sub-Saharan countries and aimed to make inferences about sub-Saharan adolescents, youths, and regions hence the random effect model was the most appropriate [45]. The proportion of COVID-19 vaccine acceptance and hesitancy was reported in pooled estimate proportion with a 95% confidence interval. In the meta-regression analysis, the coefficients were transformed by exponentiating the coefficients to obtain odds ratios. The odds ratio served as the final measure of effect and was consistently reported in the regression analysis output which was used to identify factors influencing vaccine acceptance and hesitancy in sub-Saharan African countries.

## Data analysis

We used Comprehensive Meta-Analysis (CMA) version 4 statistical software [46] to analyse the data extracted (S1 Appendix). To ensure there is consistency in the data generated from studies compiled, all data that was represented in percentages were converted into absolute numbers/or the total number of participants. When risk ratios or odds ratios were reported the primary numerical data were obtained and used. The homogeneity of the studies was assessed to determine whether they met the statistical data for meta-analysis. The confidence Intervals at 95% confidence level were calculated in logit units [47].

The random effect model was utilized for the systematic review and meta-analysis due to heterogeneity among studies from different sub-Saharan African countries ($I^2 > 50\%$), however, in cases of homogeneity during subgroup analysis, the fixed effects model was used to analyse the polled prevalence among Adolescents youths and sub-Sahara regions. The Cochrane Q-test and $I^2$ statistics were used to assess homogeneity in the study as a proportion [48, 49]. This interpretation was used because the role of $I^2$ is to provide information about the proportion of the observed variance reflecting the variance in true effects rather than sampling error [49].

To determine the group variation of COVID-19 vaccine acceptance, sub-group analysis by age groups was conducted among Adolescents and Youths of Sub-Saharan African countries this is because there are observed disparities in vaccine acceptance among age groups [50]. To determine the regional variation of COVID-19 vaccine acceptance, sub-group analysis by region was conducted because the African continent operates within the five regions out of these four (Central, eastern, western, and southern Africa) are in sub-Saharan African countries under the Africa Centres for Disease Control and Prevention under the African Union [51].

A forest plot format was used to present the pooled estimates of the prevalence of COVID-19 vaccine acceptance among adolescents and youths at 95% Confidence Intervals [52]. Sub-group analysis was done based on the study group and study region level. Publication bias was also assessed using visual inspection by using the Funnel Plot and Eggers test with p-value >0.05 indicating that there was no publication bias while the sensitivity analysis was conducted by using visual observation of the deviation of the estimates from the mean [53, 54].

Leave one out sensitivity analysis was conducted to assess the influence of a single study on the overall effect. To model the determinants of COVID-19 vaccine acceptance and hesitancy the Comprehensive Meta-Analysis (CMA) version 4 statistical software was used using Meta regression.

# Results

## Search results

A total of 887 articles were retrieved from seven (7) search engines. The number of articles retrieved from Research 4Life was 85, Google Scholar 200 manuscripts, Scopus 106, African Journal Online 220, PubMed Central (PMC) 68, PubMed 26, and Embase 150. The screening was conducted, and the removal of duplicates and records marked as ineligible by automation tools resulted in the removal of 590 articles. The titles and abstracts were screened from 297 articles, thereafter 174 records were excluded for irrelevant titles and abstracts a total of 123 articles remained. After further screening, 40 articles were excluded because the articles were not within the scope of the study. The full text of the remaining 83 were further screened 63 articles were excluded due to the study design and age group which was outside the study population. Finally, 23 articles met the study inclusion criteria (S1 Table). Fig 1, the PRISMA flowchart.

## Characteristics of included studies

A total of 23 published articles were analyzed in this review (Fig 1). These articles constituted studies on COVID-19 vaccine acceptance and hesitancy in sub-Saharan African countries (S1 Appendix). The highest number of studies were performed cross-country by Wang et al 2022 [30] (n = 4) and Faye et al., 2021 [33], (n = 4) followed by Mudenda et al [32, 35] (n = 2) and the rest of the authors had one article each.

The largest sample size (N = 998) was found in the survey conducted by Mudenda et al [32] while the smallest sample size (N = 149) was found in the survey conducted by Faye et al., [33]. Of all the 23 studies included in the study thirteen (13) studies focused on adolescents and ten (10) on youths. Furthermore, Nigeria had 4 studies followed by Ethiopia 3, with Zambia and Ghana having 2, and other countries had 1 each.

## Pooled prevalence of vaccine acceptance

The analysis is based on 23 (N = 23) studies (Fig 1). The studies in the analysis are assumed to be a random sample from studies based in Sub-Saharan countries. The pooled prevalence of vaccine acceptance was 38.7%, (0.387, CI:95%, 0.32-0.46 $I^2$ = 98%, p < 0.001) (Fig 2).

## Subgroup analysis of COVID-19 vaccine acceptance by adolescents and youths

The finding of subgroup analysis between adolescents and youths showed that the pooled prevalence of COVID-19 vaccine acceptance among adolescents was 36.1% (0.361, 95%, CI: 0.28-45, $I^2$ = 98.75%, p < 0.001). The pooled prevalence among youths was 42% (0.42, 95%, CI: 0.32–0.53 $I^2$ = 89.85%, p <0.001), (Fig 3).

## Subgroup analysis of COVID-19 vaccine acceptance by sub-Saharan regions

The finding of subgroup analysis among regions showed that the pooled prevalence of COVID-19 vaccine acceptance in the sub-Saharan African countries was high in the Western

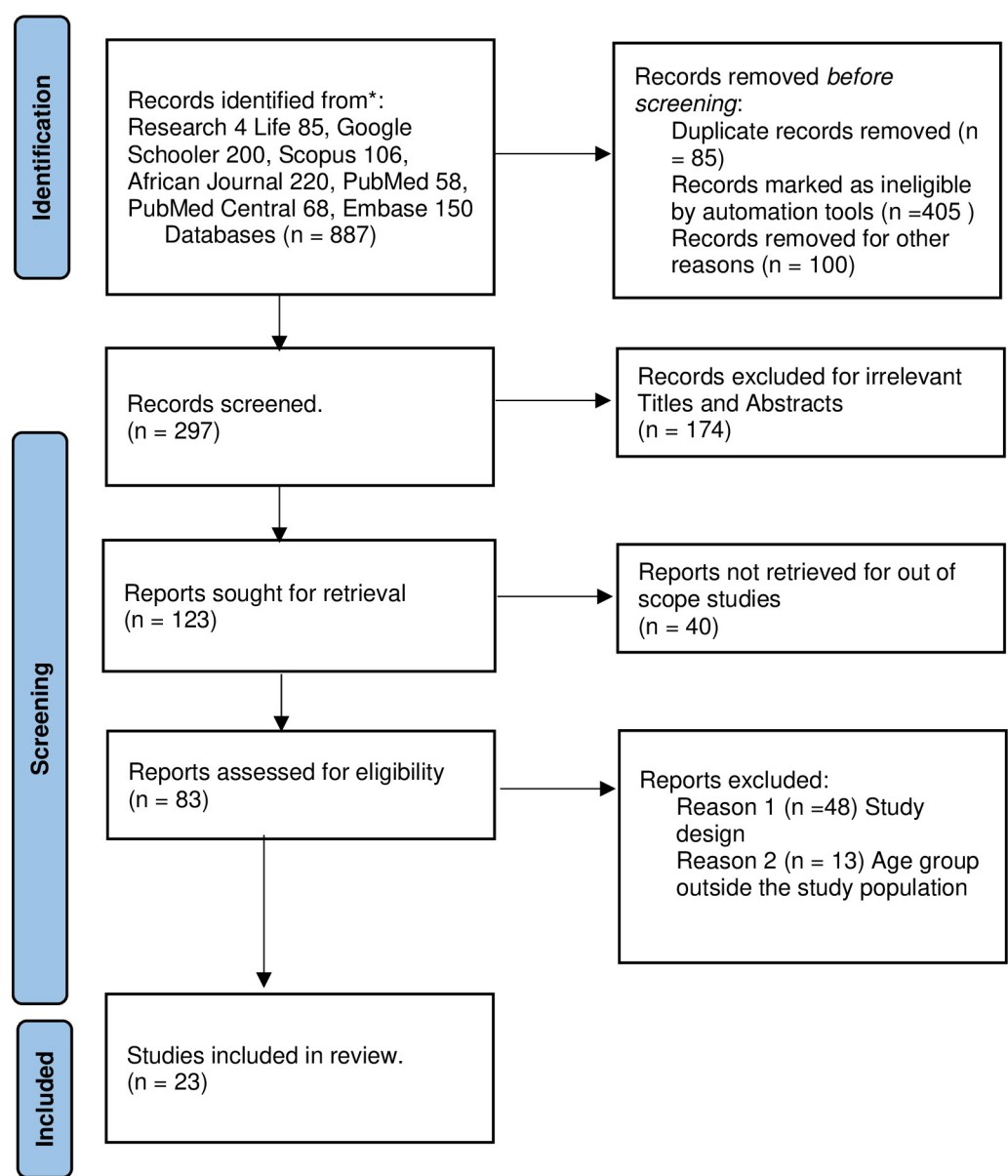

**Fig 1. Flow chart on the selection of systematic review and meta-analysis on COVID-19 vaccine acceptance, hesitancy, and its determinants in sub-Saharan African countries.**

African region, with an estimate of 42.4% (0.424, 95%, CI: 0.23-0.52 $I^2$ = 97.83%, p<0.001). Followed by Eastern Africa region, with an estimate of 39.8% (0.398, 95%, CI: 0.28-0.53, $I^2$ = 97.42%, p<0.001), Central Africa with an estimate of 33% (0.33, n = 1) (95%, CI: 0.12-0.65, $I^2$ = 0.000, p<0.001). The list was Southern Africa with an estimate of 24%, (0.24, 95%, CI: 0.129-0.407, $I^2$ = 98.48%, p< 0.001) (Fig 4).

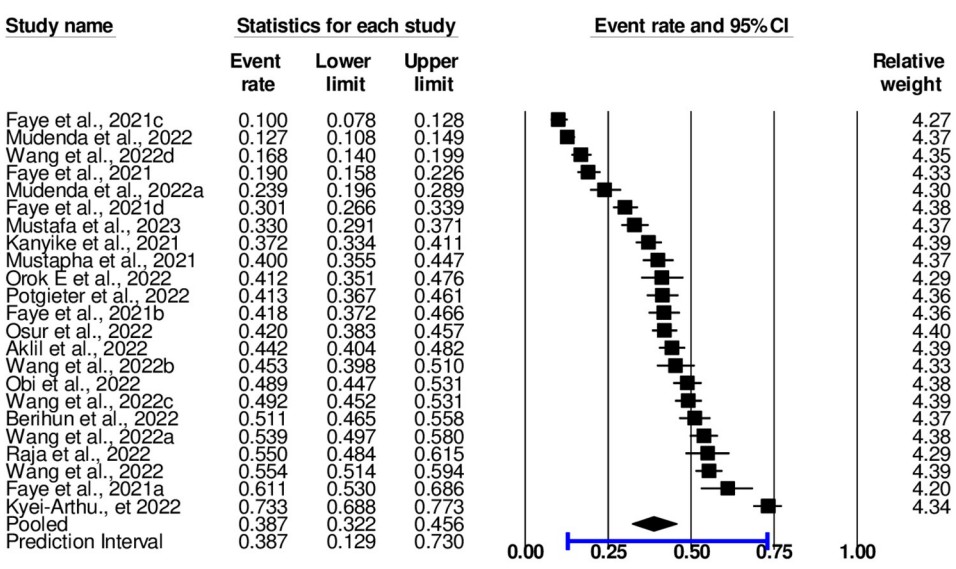

**Meta Analysis**

**Fig 2. Prevalence of vaccine acceptance in sub-Saharan African countries.**

## Study biasness and sensitivity analysis

To estimate the publication bias, we plotted the funnel plot and a sensitivity analysis of the studies. Eggar's test for publication bias was conducted with intercept -7.037, (CI, 95%, -23.40-9.33, p>0.05) indicating no study biasness (Fig 5). The studies were symmetrically distributed. Furthermore, a sensitivity analysis of the funnel plot indicated that there were no missing studies as the deviation from the mean is insignificant to show any changes to the studies (Fig 5).

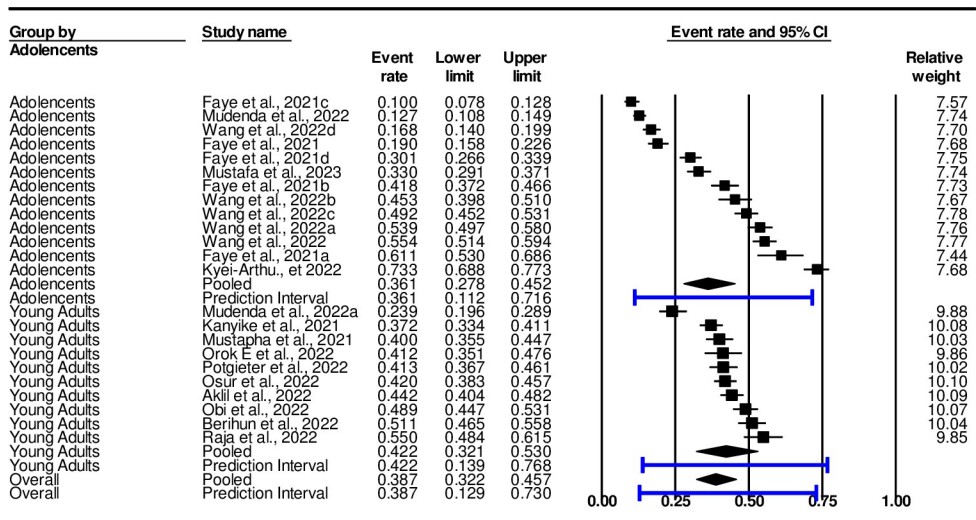

**Meta Analysis**

**Fig 3. Subgroup analysis of pooled prevalence and its determinants of adolescents and youths.**

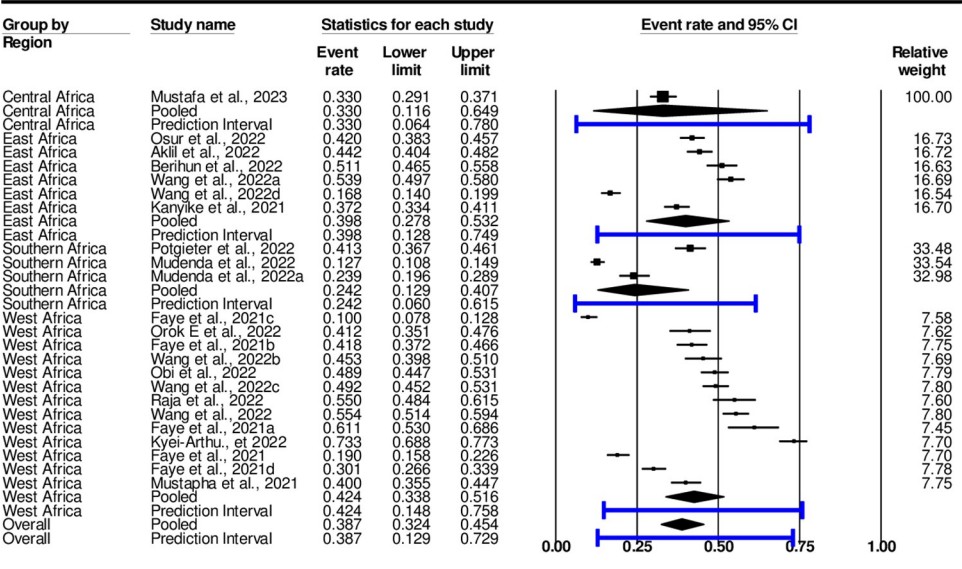

**Meta Analysis**

**Fig 4. Subgroup analysis of COVID-19 vaccine acceptance among regions.**

## Sensitivity analysis of COVID-19 vaccine acceptance among adolescents and youths

The Sensitivity Analysis: A random effect model result showed that no single study had influenced the overall pooled prevalence of COVID-19 vaccine Acceptance in sub-Saharan African countries (Fig 6). When one study was removed the results indicated that the remaining studies were within the CI of 25% and 50% (Fig 6).

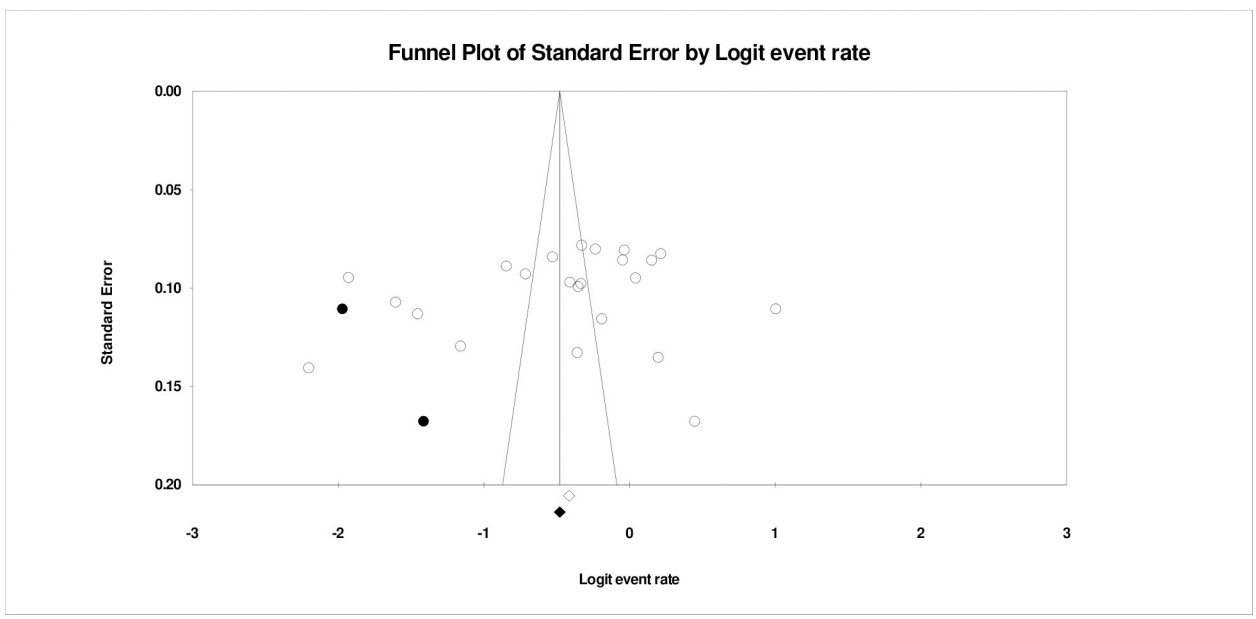

**Fig 5. Shows the study biasness analysis of the funnel plot.**

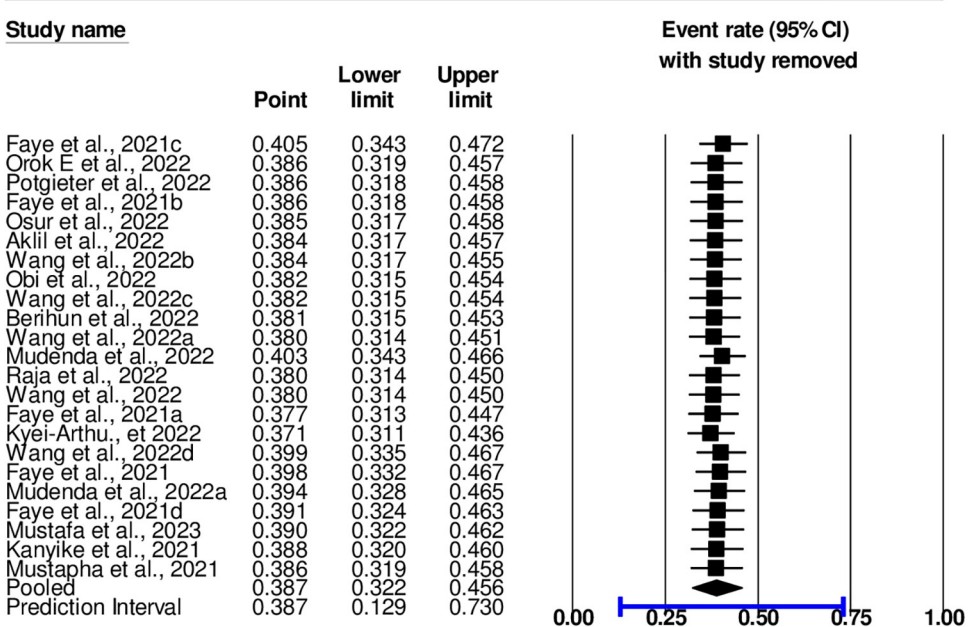

**Meta Analysis**

**Fig 6. Sensitivity analysis of COVID-19 vaccine acceptance among adolescents and youths.**

## Determinants of vaccine acceptance among adolescents and youths

The significant determinants were the odds of the desire for self-immunity had a more than 1 fold increased odds of accepting the COVID-19 vaccine (AOR =1.97, 95%, CI, 1.083.47, $I^2$ = 94.15%, p < 0.05). Receiving information from the health officer had a more than 4-folds increased odds of accepting the vaccine (AOR = 4.36, 95%, CI, 2.28-8.32, $I^2$ = 97.74, p < 0.001). Understanding the effectiveness of the COVID-19 vaccine had a more than 2-folds increased odds of accepting the vaccine (AOR = 2.14, 95%, CI, 1.14-4.05, $I^2$ = 97.4%, p < 0.05) (Table 2).

**Table 2. Determinants of COVID-19 vaccine acceptance among adolescents and youths in sub-Saharan African countries.**

| Covariate | Outcome | AOR | 95%, CI | p-value |
|---|---|---|---|---|
| Self-Immunity | No | Reference | | |
| | Yes | 1.94 | 1.08-3.47 | 0.05 |
| Health Officer Information | No | Reference | | |
| | Yes | 4.36 | 2.28-8.32 | < 0.001 |
| Effectiveness of COVID-19 vaccine | No | Reference | | |
| | Yes | 2.14 | 1.14-4.05 | 0.05 |
| Vaccine Safety | No | Reference | | |
| | Yes | 0.56 | 0.27-1.16 | 0.12 |
| Knowledge of Vaccine | No | | | |
| | Yes | 1.37 | 0.45-4.15 | 0.58 |

Ref: AOR= Adjusted Odds Ratio, CI = Confidence Interval.

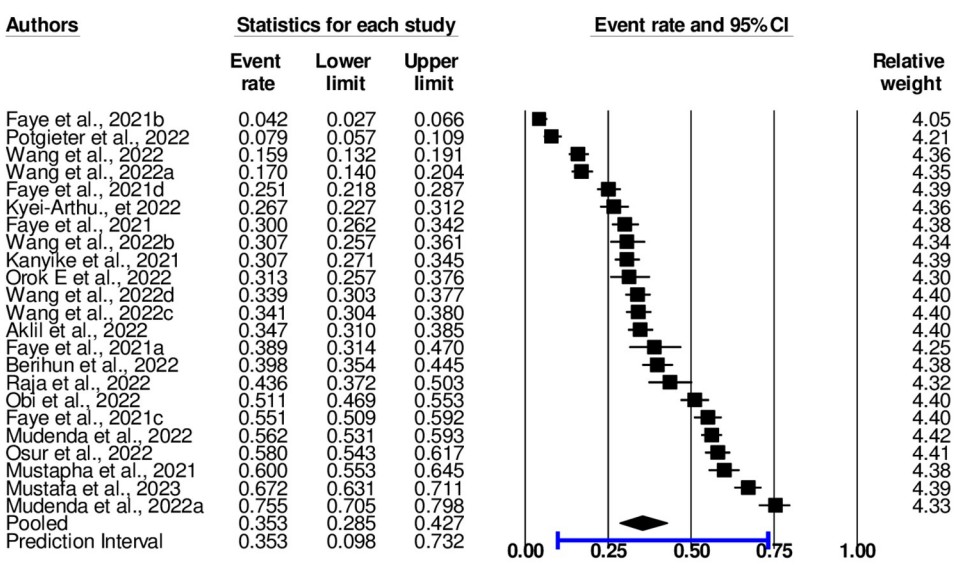

| Authors | Statistics for each study | | | Event rate and 95% CI | Relative weight |
|---|---|---|---|---|---|
| | Event rate | Lower limit | Upper limit | | |
| Faye et al., 2021b | 0.042 | 0.027 | 0.066 | | 4.05 |
| Potgieter et al., 2022 | 0.079 | 0.057 | 0.109 | | 4.21 |
| Wang et al., 2022 | 0.159 | 0.132 | 0.191 | | 4.36 |
| Wang et al., 2022a | 0.170 | 0.140 | 0.204 | | 4.35 |
| Faye et al., 2021d | 0.251 | 0.218 | 0.287 | | 4.39 |
| Kyei-Arthu., et 2022 | 0.267 | 0.227 | 0.312 | | 4.36 |
| Faye et al., 2021 | 0.300 | 0.262 | 0.342 | | 4.38 |
| Wang et al., 2022b | 0.307 | 0.257 | 0.361 | | 4.34 |
| Kanyike et al., 2021 | 0.307 | 0.271 | 0.345 | | 4.39 |
| Orok E et al., 2022 | 0.313 | 0.257 | 0.376 | | 4.30 |
| Wang et al., 2022d | 0.339 | 0.303 | 0.377 | | 4.40 |
| Wang et al., 2022c | 0.341 | 0.304 | 0.380 | | 4.40 |
| Aklil et al., 2022 | 0.347 | 0.310 | 0.385 | | 4.40 |
| Faye et al., 2021a | 0.389 | 0.314 | 0.470 | | 4.25 |
| Berihun et al., 2022 | 0.398 | 0.354 | 0.445 | | 4.38 |
| Raja et al., 2022 | 0.436 | 0.372 | 0.503 | | 4.32 |
| Obi et al., 2022 | 0.511 | 0.469 | 0.553 | | 4.40 |
| Faye et al., 2021c | 0.551 | 0.509 | 0.592 | | 4.40 |
| Mudenda et al., 2022 | 0.562 | 0.531 | 0.593 | | 4.42 |
| Osur et al., 2022 | 0.580 | 0.543 | 0.617 | | 4.41 |
| Mustapha et al., 2021 | 0.600 | 0.553 | 0.645 | | 4.38 |
| Mustafa et al., 2023 | 0.672 | 0.631 | 0.711 | | 4.39 |
| Mudenda et al., 2022a | 0.755 | 0.705 | 0.798 | | 4.33 |
| Pooled | 0.353 | 0.285 | 0.427 | | |
| Prediction Interval | 0.353 | 0.098 | 0.732 | | |

Meta Analysis

**Fig 7. Forest plot of the prevalence of COVID-19 vaccine hesitancy.**

## Pooled prevalence of vaccine hesitancy

Overall, the pooled prevalence of vaccine hesitancy in sub-Saharan African countries was estimated to be 35.3% (95%, CI 28.5-42.8, $I^2 = 98\%$, p< 0.001) (Fig 7).

## Subgroup analysis of COVID-19 vaccine hesitancy by adolescents and youths

The finding of subgroup analysis between adolescents and youths showed that the pooled prevalence of COVID-19 vaccine hesitancy among adolescents was estimated at 30.7% (0.307, n = 1395%, CI: 0.23-0.41, $I^2$ 98.46, p<0.001). While the pooled prevalence among youths was 41.6% (0.416, (95%, CI: 0.30–0.53, $I^2$ = 97.99%, p <0.001) (Fig 8).

## Subgroup analysis of COVID-19 vaccine hesitancy by regions

Overall, four regions were considered for sub-group analysis. The finding among regions showed that the pooled prevalence of COVID-19 vaccine hesitancy in the sub-Saharan African countries was high in the Central Africa region, with an estimate of 67.2% (0.672, 95%, CI: 0.32-0.9 $I^2$ = 97.421,), Southern African region was second, with an estimate of 41.7% (0.417, 95%, CI: 0.23-0.62, $I^2$ 97.64, p < 0.001). East Africa, an estimate of the pooled prevalence of 34.7% (0.347, 95%, CI: 0.23-0.49, $I^2$ = 97.71, p< 0.001) while West Africa an estimate of 31.9% (0.319, 95%, CI: 0.24-0.41, $I^2$ = 97.64, p< 0.001) (Fig 9).

## Study biasness and sensitivity analysis on studies on vaccine hesitancy

For estimating the publication bias, we plotted the funnel plot and a sensitivity analysis of the studies for vaccine hesitancy in sub-Saharan African countries. Egger's test was used to assess for study biases. Eggar's test for publication bias was conducted with intercept -14.28, CI, 95%, -25.67- (-) 2.89 p< 0.05. A sensitivity analysis of the funnel plot was conducted indicating that

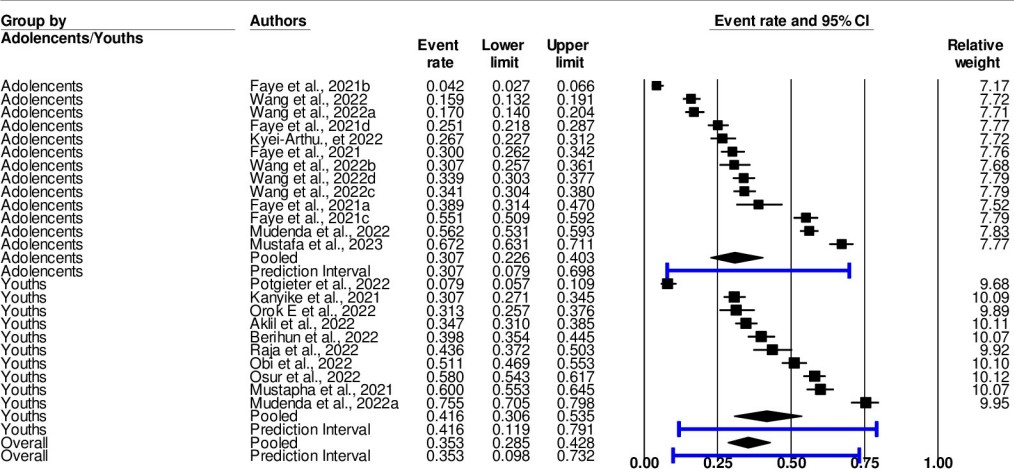

**Fig 8. Subgroup analysis of adolescents and young adults.**

there could be no missing studies identified. There was no deviation from the mean indicating that there were no missing studies (Fig 10).

## Sensitivity analysis

A random effect model result showed that no single study had influenced the overall pooled prevalence of COVID-19 vaccine Acceptance in sub-Saharan African countries (Fig 11). When one study was removed the results indicated that the remaining studies were within the PI of 25% and 50% (Fig 11).

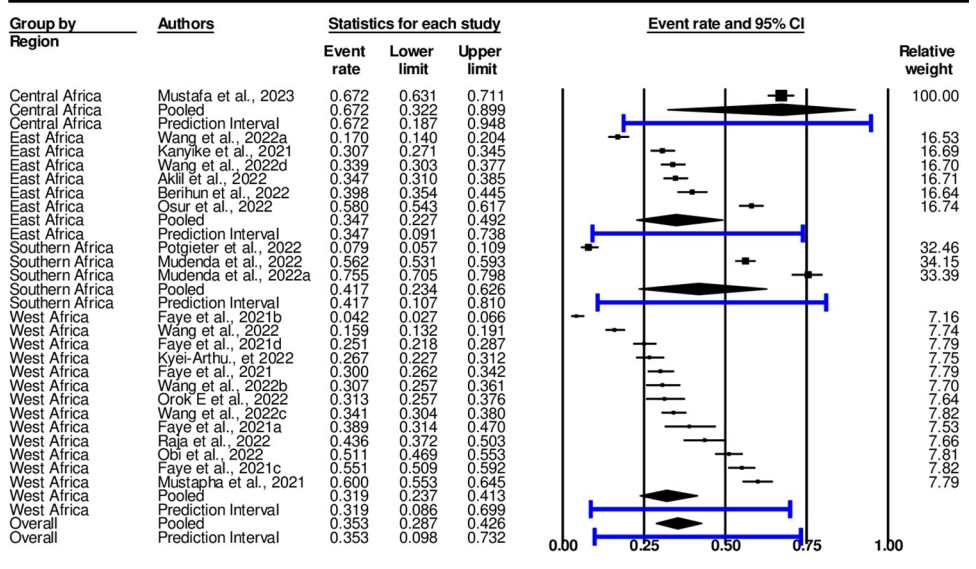

**Fig 9. COVID-19 vaccine hesitancy subgroup analysis by Sub-Saharan Region.**

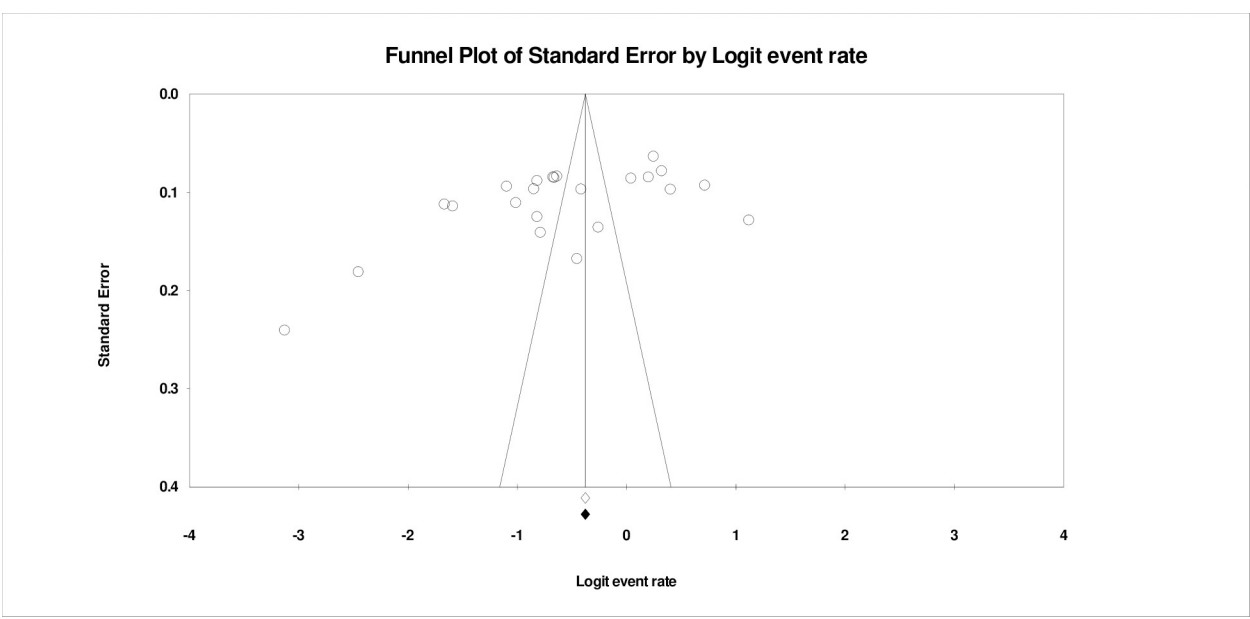

**Fig 10. Sensitivity analysis of the funnel plot for study biases.**

## Determinants of vaccine hesitancy

One factor that was identified to strongly influence vaccine hesitancy, was sources of information, to influence (Table 3). Receiving unverified sources of information on COVID-19 vaccine had a more than 1-fold increased odds of hesitancy on the vaccine. (AOR = 1.22, 95% CI, 0.10-0.45, I$^2$ = 94.09%, p < 0.001).

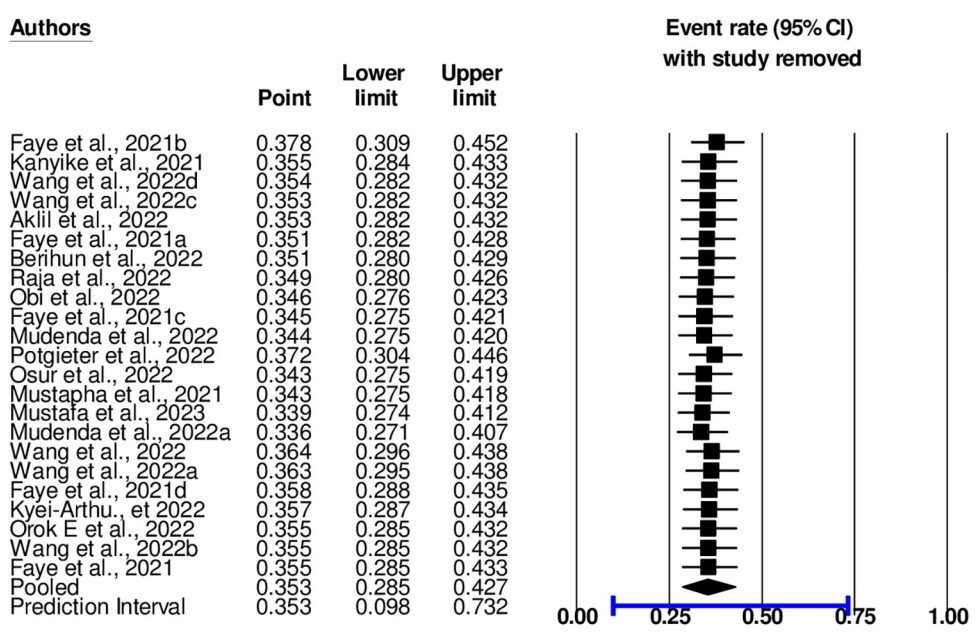

**Fig 11. Sensitivity analysis with one study removed.**

**Table 3. Determinants of COVID-19 vaccine hesitancy among adolescents and youths in sub-Saharan African countries.**

| Covariate | Outcome | AOR | 95% CI | p-value |
|---|---|---|---|---|
| Source of Information | No | Reference | | |
| | Yes | 1.22 | 0.10-1.78 | < 0.001 |
| Knowledge of COVID-19 Vaccine | No | Reference | | |
| | Yes | 2.86 | 0.84-9.68 | 0.09 |
| Effectiveness of the COVID-19 Vaccine | No | Reference | | |
| | Yes | 2.61 | 0.90-7.54 | 0.08 |
| Vaccine Safety | No | | | |
| | Yes | 0.07 | 0.001-4.81 | 0.22 |

AOR= Adjusted Odds Ratio, CI = Confidence Interval

## Discussion

The review investigated the polled prevalence of COVID-19 vaccine acceptance, hesitancy, and its determinants among adolescents and youths in sub-Saharan African countries. According to the findings, the prevalence of vaccine acceptance in sub-Saharan African countries is low with a pooled prevalence of 38.7%, while hesitancy was estimated to be at 35.3%. These findings are similar to what Wang et al., [30] and Mudenda et al, [53] identified that the level of vaccine acceptance and hesitancy was alarming in most of the sub-Saharan African countries because they are low.

In this study, the pooled prevalence among adolescents of vaccine acceptance was at 36.1% while hesitancy was at 30.7%, which is low compared to other regions such as USA with a polled prevalence of 52%, Europe 55% and Asia 72% [54]. Furthermore, our findings indicate that the pooled prevalence of vaccine acceptance among youths was estimated to be 42.2% while hesitancy was estimated at 41.6% these results demonstrate to be lower than in a study conducted in Stockholm which had 65.7% uptake among youths [55]. The overall findings of the results obtained from sub-Saharan African countries among adolescents and youths are below the requirement to achieve herd immunity against COVID-19, which requires that a substantial proportion of the population be vaccinated to lower the infection rate in the population [56].

At the regional level, the pooled prevalence of vaccine acceptance was lowest in central Africa at 33% acceptance while hesitancy with an estimated 67.2%. Though a study by Miner [57] indicated that the level of vaccine acceptance is high among Sub-Saharan African Countries the findings of this study have demonstrated a low level of COVID-19 vaccine acceptance among adolescents and youths compared to other regions as demonstrated by other studies [54, 55].

Southern Africa with the pooled prevalence of acceptance at 24.2% and hesitancy estimated at 41.7% still indicates the need for improved vaccine acceptance similarly studies conducted in the general population indicate low acceptance of COVID-19 vaccine and high level of hesitancy in this region [31].

East Africa had an acceptance rate of 39.8% and an estimated hesitancy rate of 34.7%. Similar, to the findings of Kyei-Arthur [31] vaccine hesitancy levels in East Africa are still low, slightly higher than in Southern Africa among adolescents and youths. West Africa had the highest rates among sub-Saharan regions with an acceptance rate of 42.4%, and an estimated hesitancy rate of 31.9%. The findings of this study align with those of Wang et al., [30] indicating that the level of vaccine hesitancy is still high in the region although it is the lowest among all Sub-Saharan African countries. Other studies have reported similar levels of vaccine acceptance and hesitancy in West African countries [33, 36, 38].

Therefore, to completely end the pandemic, there is a need to increase vaccine uptake and reduce hesitancy among adolescents and youths to eradicate COVID-19 disease [58]. This calls for the need to adequately understand each population receiving the vaccine which can not be emphasised enough [59].

The regression analysis results revealed that health workers' information was found to have a positive influence on COVID-19 vaccination similar to the findings of Ojo et al., [60] who found that the influence of health workers and government media departments had a much greater impact on influencing COVID-19 vaccine uptake. The trust in the effectiveness of the vaccine against COVID-19 and the desire to develop immunity had influenced an increase in vaccine acceptance among adolescents and youths similar to the findings of Roy et al., [61] who indicated that acceptance of the COVID-19 vaccine included confidence in vaccine safety and its effectiveness. These results are similar to the findings of Amo-Adjei [62] in Ghana, who indicated that trust and the effectiveness of the COVID-19 vaccine, influenced individuals to get vaccinated from a range of moderate to high vaccination rates. Similarly, the findings of Faye indicate that the perceived effectiveness and safety of COVID-19 vaccines increased willingness to get vaccinated.

Different sources of information influenced increasing vaccine hesitancy similar to the findings of Osuagwu et al., [63], who found that all six information sources such as social media, television, health workers, family, and friends were strongly associated with vaccine hesitancy in sub-Saharan African countries, except radio was not a strong predictor. The findings are also similar to those of Ganem et al., [64] in the USA who found that the key reasons to not vaccinate themselves, or their children, were concern about side effects, insufficient research about the effect of the vaccine in children, the rapid development of vaccines, the necessity for more information and previous infection by COVID-19.

Knowledge of the COVID-19 vaccine and its effectiveness was not associated with vaccine acceptance or hesitancy contrary to the findings of studies by Orok in Nigeria [42] who indicated that knowledge and perception of COVID-19 vaccines can improve vaccine acceptance. Our findings further indicate that the effectiveness of the vaccine and safety did not have an influence on hesitancy among youths and adolescents in sub-Saharan African countries. However, the findings in Malaysia by Alwi et al., [65] in the general population indicated that despite the findings of a high acceptance rate of COVID-19 vaccine effectiveness and safety were still key determinants to continue to be sensitizing the public to improve acceptance. Source of information as highlighted in the results had a positive influence on increasing hesitancy according to the findings of Kashte et al [66] and Hong et al, [67] there is a great need to encourage the development of public trust in official media through rapid dissemination of information in a transparency manner explaining the rapid development of the COVID-19 vaccine. Furthermore, initiatives such as COVID-19 champions increasing awareness among youths aged 18 to 34 years should be encouraged [68].

While the global risk assessment of COVID-19 remains high, there is a need to ensure that the recommendation by the WHO of integrating the vaccination program into life course programs is implemented [69]. The need for countries to put concerted efforts to increase COVID-19 vaccination coverage for all people in the high-priority groups while reducing hesitancy (as defined by the SAGE Roadmap of April 2023) [69].

## Conclusion

Based on the findings of the study the prevalence of vaccine acceptance among adolescents and youths in sub-Saharan African countries is low across the four regions of sub-Saharan African countries. Determinants of vaccine acceptance and hesitancy tend to differ by category

of age and region country in Sub-Saharan African countries. Therefore, there is a need to ensure that extensive research is undertaken into age-appropriate health promotion messages and strategies to encourage uptake of the vaccines in the populations. This will reduce the transmission of infectious diseases among adolescents and youths who are the majority in the population. Educating individuals on the benefits of vaccination and the expected side effects should be addressed during the vaccination rollout. Furthermore, policymakers should design education programs that explain the contemporary process of vaccine production to the masses to reduce hesitancy among members of the public.

## Limitations of the study

The strength of the study is the use of the PRISMA flow charts, and five databases were searched. However, this study is not void of limitations. The web searches were restricted only to English language, reporting bias was evident. Second, relevant predictors might be missed. Furthermore, the studies extracted had a different measure of the outcome as a result the study used the role data and converted it to invent rate across all the studies to obtain proportions.

Not all studies reported odds ratios. The absence of studies in some countries and regions would raise questions about the generalisation of the study findings.

## Supporting information

**S1 Table. Study characteristics included in the systematic review and meta-analysis on the prevalence of COVID-19 vaccine acceptance, hesitancy, and its determinants in sub-Saharan African countries.**
(DOCX)

**S1 Appendix. Suplemnatry material as data used in the systematic review and meta-analysis on the prevalence of COVID-19 vaccine acceptance, hesitancy, and its determinants in sub-Saharan African countries.**
(XLSX)

**S1 File.**
(PDF)

## Author Contributions

**Conceptualization:** Allan Mayaba Mwiinde.

**Data curation:** Allan Mayaba Mwiinde.

**Formal analysis:** Allan Mayaba Mwiinde.

**Funding acquisition:** Joseph Mumba Zulu.

**Investigation:** Allan Mayaba Mwiinde.

**Methodology:** Allan Mayaba Mwiinde, Isaac Fwemba.

**Project administration:** Allan Mayaba Mwiinde, Joseph Mumba Zulu.

**Software:** Allan Mayaba Mwiinde.

**Supervision:** Patrick Kaonga, Choolwe Jacobs, Joseph Mumba Zulu, Isaac Fwemba.

**Validation:** Allan Mayaba Mwiinde, Patrick Kaonga, Choolwe Jacobs, Joseph Mumba Zulu, Isaac Fwemba.

**Visualization:** Allan Mayaba Mwiinde, Patrick Kaonga, Choolwe Jacobs, Isaac Fwemba.

**Writing – original draft:** Allan Mayaba Mwiinde.

**Writing – review & editing:** Allan Mayaba Mwiinde, Patrick Kaonga, Choolwe Jacobs, Joseph Mumba Zulu, Isaac Fwemba.

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
