## [Decision Letter · Decision Letter 0]

8 Aug 2024

PONE-D-24-09721Determinants of COVID-19 Vaccine Acceptance and Hesitance among Adolescents and Youths aged 10-35 years in Sub-Saharan African Countries: A Systematic Review and Meta-analysisPLOS ONE

Dear Dr. Mwiinde,

Thank you for submitting your manuscript to PLOS ONE. After careful consideration, we feel that it has merit but does not fully meet PLOS ONE’s publication criteria as it currently stands. Therefore, we invite you to submit a revised version of the manuscript that addresses the points raised during the review process.

We look forward to receiving your revised manuscript.

Kind regards,

Richard Makurumidze (MBChB, MPH, PhD)

Academic Editor

PLOS ONE

Reviewers' comments:

Reviewer's Responses to Questions

**Comments to the Author**

1. Is the manuscript technically sound, and do the data support the conclusions?

Reviewer #1: Yes

Reviewer #2: Yes

Reviewer #3: Yes

2. Has the statistical analysis been performed appropriately and rigorously? 

Reviewer #1: Yes

Reviewer #2: Yes

Reviewer #3: Yes

3. Have the authors made all data underlying the findings in their manuscript fully available?

Reviewer #1: Yes

Reviewer #2: Yes

Reviewer #3: No

4. Is the manuscript presented in an intelligible fashion and written in standard English?

Reviewer #1: Yes

Reviewer #2: Yes

Reviewer #3: Yes

5. Review Comments to the Author

Reviewer #1: In the limitations section, could the fact that all the included studies were cross-sectional also be a limitation?

Please, correct the term hesitance to hesitancy throughout the manuscript.

I look forward to seeing the published version of this manuscript

Reviewer #2: Address the following information in the discussion and introduction section:

Impact of misinformation and disinformation about COVID-19 vaccines on public perception and acceptance

Healthcare providers, government agencies, and pharmaceutical companies role in vaccine acceptance.

Long-Term Impact for sustained acceptance

Reviewer #3: Thank you for giving me an opportunity to review this article about “Determinants of COVID-19 Vaccine Acceptance and Hesitance among Adolescents and Youths aged 10-35 years in Sub-Saharan African Countries: A Systematic Review and Meta-analysis.”

Introduction

It is not so critical as to why this study among adolescents and youth is essential? Is it a COVID-19 high risk group? Why should we worry about them? This could be fixed in the last paragraph

Looks like citations 17 to 20 in the second last paragraph have been manually inserted. Please rectify this.

In the statement, “As of February 2024, the percentage of COVID-19 vaccination in Africa was at 51.85…” is that a percentage?

Methodology

Though the inclusion criteria look fine, the authors just define some of the exclusion criteria as the opposite of inclusion which seems not right. I would advise the authors to use the PECO framework by Morgan RL (2018) to define eligibility.

Proof read the information sources section and remove any repetitions made.

The search strategy is not clear. Is what is being presented the full string that was used? And in which database was this used? Normally, a search is built in PubMed and modified for other databases. Additionally, it would be very useful to present the full search strategy indicating the strings as search in each of the databases together with the number of records and any filters added.

An important step of study selection is mixed-up; this makes following the article hard. I would advise the authors to be systematic following the five steps of systematic reviews clearly. Question, Search, Screening, Extraction, quality assessment, Data management. This flow would make readability easy.

Did you use Endnote, Rayyan of Covidence for screening? please be specific with this. Like I have noted above, please rearrange the write up for logical flow.

In the data analysis section, did you consider heterogeneity at 12 of 50%, or 75%. Please harmonize this.

Results

In the characteristics of included studies, it would be logical to state the reasons to why some studies were excluded after being subjected to full-text screening. this should also include how many studies were excluded for which reason.

It is also not clear the basis on which you performed the subgroup analysis. Please make this clear.

In Figure 1, please state the reasons for exclusion, not just reason 1 and reason 2

For verification purposes, please present a table for study characteristics indicating all the 23 studies, well cited with a column for quality of included studies.

Give keen attention to your P-values most especially in the subgroup analysis by region. First of all, it is a best practice to write P values with three digits after the decimal point (e.g. P=0.056), secondly, any p value that is displayed as 0.000 is written as p<0.001. finally, some of your p values have a greater than sign instead of less than sign, please rectify this.

In the determinants of vaccination acceptance section, I am still confused by the statement, “The odds of receiving Health Officers' information were 4.36 times more likely to influence acceptance of the COVID-19 vaccine than those who said no to the information from the health workers (AOR=4.36, 95%, CI, 2.28-8.32, I2=97.74, p < 0.001).” where the odds more likely? Or those who reported to have received the information were more likely? Please revise how you make these statement. The easiest way would have been, receiving information from the Health officer increased the odds of acceptance of the vaccine by more than 4-folds, or Receiving information from the health officer had a more than 4-folds increased odds of accepting the vaccine.

On your Tables, put a footnote to describe what AOR and CI mean.

Discussion

The first paragraph of your discussion seems to be more fitting into the introduction section.

The findings relating to the main outcome seem not well discussed. No adequate comparisons, no implications, I think you need to improve on this.

6. PLOS authors have the option to publish the peer review history of their article (what does this mean?). If published, this will include your full peer review and any attached files.

Reviewer #1: **Yes: **Dr Sahabi Kabir Kabir Sulaiman

Reviewer #2: No

Reviewer #3: No

---

## [Author Response · Author response to Decision Letter 0]

29 Aug 2024

Thank you to the reviewers for the constructive feedback on our manuscript entitled “Determinants of COVID-19 Vaccine Acceptance and Hesitancy among Adolescents and Youths aged 10-35 years in Sub-Saharan African Countries: A Systematic Review and Meta-analysis.” We have addressed point by point each reviewer's comments and concerns, and we hope this will improve the manuscript. 

Introduction

Comment: It is not so critical as to why this study among adolescents and youth is essential? Is it a COVID-19 high risk group? Why should we worry about them? This could be fixed in the last paragraph

Response: We wish to thank the reviewer for the comment. The statement has been included in the last paragraph as advised as follows; “This is because adolescents and youths have a high risk of COVID-19 infection and the potential for high transmission rate in the communities, therefore, causing continued circulation of the virus within the population [27]”.

Comment: Looks like citations 17 to 20 in the second last paragraph have been manually inserted. Please rectify this. 

Response: Citations 17,19,20 have been rectified in accordance with the advice given. 

Comment: In the statement, “As of February 2024, the percentage of COVID-19 vaccination in Africa was at 51.85…” is that a percentage?

Response: We wish to thank the reviewer for the guidance. The proportion has been included as advised. “The percentage has been inserted as follows 51.85%”. 

Methodology

Comment: Though the inclusion criteria look fine, the authors just define some of the exclusion criteria as the opposite of inclusion which seems not right. I would advise the authors to use the PECO framework by Morgan RL (2018) to define eligibility.

We wish to thank the reviewer for the guidance provided. The Exclusion criteria has been re-written as follows: “Abstracts, articles, without full text, systematic reviews, conference papers, editorial letters, protocols, program evaluation report, qualitative studies, programme evaluation reports, and studies that were considered with low quality in the assessment were excluded from the study”. However, we did not use the PECO frame work by Morgan et al,. 2018. 

Comment: Proof read the information sources section and remove any repetitions made.

Response: We thank the reviewer for the guidance. The information has been proofread and reputations of the database have been removed and re-arranged in alphabetical order as follows: “A literature search was conducted using African, Journal Online, Embase, Google Schooler, PubMed Central, Research 4 Life, Scopus were used to search for the literature to be extracted”.

Comment: The search strategy is not clear. Is what is being presented the full string that was used? And in which database was this used? Normally, a search is built in PubMed and modified for other databases. Additionally, it would be very useful to present the full search strategy indicating the strings as search in each of the databases together with the number of records and any filters added.

Response: We wish to thank the reviewer for the comment we have included a statement which states that “The search string was developed using “AND” and “OR” Boolean operators”. The number of records have been indicated in the results section of the manuscript. We have also included the main search strategy that was used in Pubmed which is as follows: ”: (((((((((((((((COVID-19) OR (Corona Virus)) OR (SARS COV-2)) OR (Severe Acute Respiratory Syndrome)) AND (Vaccine)) OR (Immunization)) OR (Covid-19 Vaccine)) AND (Acceptance)) OR (Willingness)) OR (Intention)) OR (Uptake)) AND (Adolescents)) AND (Young Adults)) OR (Youths)) AND (Sub-Saharan Africa)) OR (African Countries). We wish to inform the reviewer that the full search strategy which is too long to be included in the manuscript was also done: However, the guidance from the Journals is to have the key search terms used in the manuscript. The full search strategy is as follows; (((((((("covid 19"[All Fields] OR "covid19"[All Fields] OR "covid 19"[MeSH Terms] OR "covid 19 vaccines"[All Fields] OR "covid 19 vaccines"[MeSH Terms] OR "covid 19 serotherapy"[All Fields] OR "covid 19 serotherapy"[MeSH Terms] OR "covid 19 nucleic acid testing"[All Fields] OR "covid 19 nucleic acid testing"[MeSH Terms] OR "covid 19 serological testing"[All Fields] OR "covid 19 serological testing"[MeSH Terms] OR "covid 19 testing"[All Fields] OR "covid 19 testing"[MeSH Terms] OR "sars cov 2"[All Fields] OR "sarscov2"[All Fields] OR "sarscov 2"[All Fields] OR "sars cov2"[All Fields] OR "sars cov 2"[MeSH Terms] OR "severe acute respiratory syndrome coronavirus 2"[All Fields] OR "2019 ncov"[All Fields] OR (("coronavirus"[MeSH Terms] OR "coronavirus"[All Fields] OR "cov"[All Fields] OR "ncov"[All Fields]) AND 2019/11/01:3000/12/31[Date - Publication]) OR (("corona"[All Fields] OR "coronae"[All Fields] OR "coronas"[All Fields]) AND ("virology"[MeSH Subheading] OR "virology"[All Fields] OR "viruses"[All Fields] OR "viruses"[MeSH Terms] OR "virus s"[All Fields] OR "viruse"[All Fields] OR "virus"[All Fields])) OR ("sars cov 2"[MeSH Terms] OR "sars cov 2"[All Fields] OR "sars cov 2"[All Fields]) OR ("severe acute respiratory syndrome"[MeSH Terms] OR ("severe"[All Fields] AND "acute"[All Fields] AND "respiratory"[All Fields] AND "syndrome"[All Fields]) OR "severe acute respiratory syndrome"[All Fields])) AND ("vaccin"[Supplementary Concept] OR "vaccin"[All Fields] OR "vaccination"[MeSH Terms] OR "vaccination"[All Fields] OR "vaccinable"[All Fields] OR "vaccinal"[All Fields] OR "vaccinate"[All Fields] OR "vaccinated"[All Fields] OR "vaccinates"[All Fields] OR "vaccinating"[All Fields] OR "vaccinations"[All Fields] OR "vaccination s"[All Fields] OR "vaccinator"[All Fields] OR "vaccinators"[All Fields] OR "vaccine s"[All Fields] OR "vaccined"[All Fields] OR "vaccines"[MeSH Terms] OR "vaccines"[All Fields] OR "vaccine"[All Fields] OR "vaccins"[All Fields])) OR ("immune"[All Fields] OR "immuned"[All Fields] OR "immunes"[All Fields] OR "immunisation"[All Fields] OR "vaccination"[MeSH Terms] OR "vaccination"[All Fields] OR "immunization"[All Fields] OR "immunization"[MeSH Terms] OR "immunisations"[All Fields] OR "immunizations"[All Fields] OR "immunise"[All Fields] OR "immunised"[All Fields] OR "immuniser"[All Fields] OR "immunisers"[All Fields] OR "immunising"[All Fields] OR "immunities"[All Fields] OR "immunity"[MeSH Terms] OR "immunity"[All Fields] OR "immunization s"[All Fields] OR "immunize"[All Fields] OR "immunized"[All Fields] OR "immunizer"[All Fields] OR "immunizers"[All Fields] OR "immunizes"[All Fields] OR "immunizing"[All Fields]) OR ("covid 19 vaccines"[MeSH Terms] OR ("covid 19"[All Fields] AND "vaccines"[All Fields]) OR "covid 19 vaccines"[All Fields] OR "covid 19 vaccine"[All Fields])) AND ("accept"[All Fields] OR "acceptabilities"[All Fields] OR "acceptability"[All Fields] OR "acceptable"[All Fields] OR "acceptably"[All Fields] OR "acceptance"[All Fields] OR "acceptances"[All Fields] OR "acceptation"[All Fields] OR "accepted"[All Fields] OR "accepter"[All Fields] OR "accepters"[All Fields] OR "accepting"[All Fields] OR "accepts"[All Fields])) OR "Willingness"[All Fields] OR ("intention"[MeSH Terms] OR "intention"[All Fields] OR "intent"[All Fields] OR "intentions"[All Fields] OR "intentional"[All Fields] OR "intentioned"[All Fields] OR "intents"[All Fields]) OR (("uptake"[All Fields] OR "uptakes"[All Fields] OR "uptaking"[All Fields]) AND ("covid 19 vaccines"[MeSH Terms] OR ("covid 19"[All Fields] AND "vaccines"[All Fields]) OR "covid 19 vaccines"[All Fields] OR "covid 19 vaccine"[All Fields]))) AND ("adolescences"[All Fields] OR "adolescency"[All Fields] OR "adolescent"[MeSH Terms] OR "adolescent"[All Fields] OR "adolescence"[All Fields] OR "adolescents"[All Fields] OR "adolescent s"[All Fields]) AND ("young adult"[MeSH Terms] OR ("young"[All Fields] AND "adult"[All Fields]) OR "young adult"[All Fields] OR ("young"[All Fields] AND "adults"[All Fields]) OR "young adults"[All Fields])) OR ("adolescent"[MeSH Terms] OR "adolescent"[All Fields] OR "youth"[All Fields] OR "youths"[All Fields] OR "youth s"[All Fields])) AND ("africa south of the sahara"[MeSH Terms] OR ("africa"[All Fields] AND "south"[All Fields] AND "sahara"[All Fields]) OR "africa south of the sahara"[All Fields] OR ("sub"[All Fields] AND "saharan"[All Fields] AND "africa"[All Fields]) OR "sub saharan africa"[All Fields])) OR (("african people"[MeSH Terms] OR ("african"[All Fields] AND "people"[All Fields]) OR "african people"[All Fields] OR "africans"[All Fields] OR "black people"[MeSH Terms] OR ("black"[All Fields] AND "people"[All Fields]) OR "black people"[All Fields] OR "african"[All Fields]) AND ("countries"[All Fields] OR "country"[All Fields] OR "country s"[All Fields] OR "country’s"[All Fields])).

Comment: An important step of study selection is mixed-up; this makes following the article hard. I would advise the authors to be systematic following the five steps of systematic reviews clearly. Question, Search, Screening, Extraction, quality assessment, Data management. This flow would make readability easy.

Response: We thank the reviewer for the guidance. We have included the research questions as follows; The research questions considered for this systematic review included the following. What is the prevalence of COVID-19 vaccine acceptance and hesitancy among adolescents and youths in Sub-Saharan African countries? What is the extent of variation in the COVID-19 vaccine’s acceptance and hesitancy rate among adolescents and youths in the sub-Saharan regions? What are the determinants of COVID-19 vaccine acceptance and hesitancy among adolescents and youths in sub-Saharan African Countries? We have also followed the flow as guided for the review and meta-analysis as follows, (1) Research Question (2) Outcome measurement (3) Information sources (4) Search Strategy (5) Study Screening (6) Selection process (7) Quality assessment (8) Data Collection Process and Risk of Bias (9) Measure of Effect and (10) Data Analysis. 

Comment: Did you use Endnote, Rayyan of Covidence for screening? please be specific with this. Like I have noted above, please rearrange the write up for logical flow.

Response: We thank the reviewer for the guidance, we did not use any of the above tool for initial screening. The screening was done manually using two independent researchers. 

In the data analysis section, did you consider heterogeneity at 12 of 50%, or 75%. Please harmonize this. 

We wish to thank the reviewer for the comment. The Cochrane Q-test and I2 statistics were used to assess homogeneity in the study as a proportion. This interpretation was used because the role of I2 is to provide information about the proportion of the observed variance reflecting the variance in true effects rather than sampling error [50]. P 7. 

Therefore, we are of the view that I2 or heterogeneity is not determined at 50% or 75% according to the recently published literature contrary to the interpretations by Haggins et al., 2017. The following is the extract of the interpretation in full from the cited published manuscript among many others by Borenstein et al. 2017. “Additionally, there are widely used benchmarks for I2 For example, I2 values of 25%, 50%, and 75% have been interpreted as representing small, moderate, and high levels of heterogeneity. These are seen to provide a convenient context for discussing the results of any analysis. For these reasons, the use of I2 as the primary basis for discussing how much heterogeneity is present and the use of benchmarks for interpreting the magnitude of heterogeneity have become ubiquitous in meta-analysis. Unfortunately, the use of I2 in this way is inappropriate. It represents a fundamental misunderstanding of what I2 is and how it should (and should not) be used”. Reference, Borenstein M, Higgins JP, Hedges LV, & Rothstein HR. Basics of meta-analysis: I2 is not an absolute measure of heterogeneity. Res Synth Methods. 2017; 8(1), 5-18.

Response: We wish to thank the reviewer for the comment. We wish to inform the reviewer that heterogeneity was not considered as a single proportion in a study it was considered from low, moderate, and high as indicated in the methodology “The assessment was based on if I2 ranges between 0 and 100% with values of 0–25% demonstrating low heterogeneity, 26 – 74% moderate heterogeneity and 75–100% illustrating considerable heterogeneity”. The results of the research can fall within the proportions indicated [50].

Results

Comment: In the characteristics of included studies, it would be logical to state the reasons to why some studies were excluded after being subjected to full-text screening. This should also include how many studies were excluded for which reason.

Response: Search Results 

A total of 887 articles were retrieved from seven (7) search engines. The number of articles retrieved from Research 4Life was 85, Google Scholar 200 manuscripts, Scopus 106, African Journal Online 220, PubMed Central (PMC) 68, PubMed 26, and Embase 150. The screening was conducted, and the removal of duplicates and records marked as ineligible by automation tools resulted in the removal of 590 articles. The titles and abstracts were screened from 297 articles, thereafter 174 records were excluded for irrelevant titles and irrelevant abstract and a total of 123 articles remained. After further screening, 40 articles were excluded because the articles were not within the scope of the study. The full text of the remaining 83 were further screened 63 articles were excluded due to the study design and study areas. Finally, 23 articles met the study inclusion criteria. 

Comment: It is also not clear the basis on which you performed the subgroup analysis. Please make this clear.

Response: We thank the reviewer for the comment we have included the justification for all the sub-group analysis in the methodology section we hope that the justification fits better under methodology; “To determine the group variation of COVID-19 vaccine acceptance, sub-group analysis by age groups was conducted among Adolescents and Youths of Sub-Saharan African countries this is because there are observed disparities in vaccine acceptance among age groups [51]. To determine the regional variation of COVID-19 vaccine acceptance, sub-group analysis by region was conducted this is because the African continent operates within the five regions under the Africa Centres for Disease Control and Prevention under the African Union, out of these four (Central, eastern, western, and southern Africa) are in sub-Saharan African countries [52]. .

Comment: In Figure 1, please state the reasons for exclusion, not just reason 1 and reason 2.

Response: We thank the reviewer for the comment we have included the reasons in Figure 1 as guided “1. Study design 2. Age group was outside the study population”.

Comment: For verification purposes, please present a table for study characteristics indicating all the 23 studies, well cited with a column for quality of included studies.

Response: We sincerely appreciate the comment however, we wish to inform the reviewer that the table was attached with the manuscript submission as S-1 Appendix: Study characteristics included in the systematic review and meta-analysis on the prevalence of COVID-19 vaccine acceptance and its determinants in sub-Saharan African countries.

Comment: Give keen attention to your P-values most especially in the subgroup analysis by region. First of all, it is a best practice to write P values with three digits after the decimal point (e.g. P=0.056), secondly, any p value that is displayed as 0.000 is written as p<0.001. finally, some of your p values have a greater than sign instead of less than sign, please rectify this.

Response: We wish to thank the reviewer for the guidance provided. All p values have been amended and written in accordance with the standard statistical procedure e.g “The finding of subgroup analysis between adolescents and youths showed that the pooled prevalence of COVID-19 vaccine acceptance among adolescents was 36.1% (0.361, 95%, CI: 0.28-45, I2= 98.75%, p < 0.001). The pooled prevalence among youths was 42% (0.42, 95%, CI: 0.32 - 0.53 I2 = 89.85%, p <0.001)”. further

---

## [Decision Letter · Decision Letter 1]

8 Sep 2024

Determinants of COVID-19 Vaccine Acceptance and Hesitancy among Adolescents and Youths aged 10-35 years in Sub-Saharan African Countries: A Systematic Review and Meta-analysis

PONE-D-24-09721R1

Dear Dr. Mwiinde,

We’re pleased to inform you that your manuscript has been judged scientifically suitable for publication and will be formally accepted for publication once it meets all outstanding technical requirements.

Kind regards,

Dr Richard Makurumidze (MBChB, MPH, PhD)

Academic Editor

PLOS ONE

Additional Editor Comments (optional):

Reviewers' comments:

Reviewer's Responses to Questions

**Comments to the Author**

1. If the authors have adequately addressed your comments raised in a previous round of review and you feel that this manuscript is now acceptable for publication, you may indicate that here to bypass the “Comments to the Author” section, enter your conflict of interest statement in the “Confidential to Editor” section, and submit your "Accept" recommendation.

Reviewer #1: All comments have been addressed

Reviewer #3: (No Response)

2. Is the manuscript technically sound, and do the data support the conclusions?

Reviewer #1: Yes

Reviewer #3: (No Response)

3. Has the statistical analysis been performed appropriately and rigorously? 

Reviewer #1: Yes

Reviewer #3: (No Response)

4. Have the authors made all data underlying the findings in their manuscript fully available?

Reviewer #1: Yes

Reviewer #3: (No Response)

5. Is the manuscript presented in an intelligible fashion and written in standard English?

Reviewer #1: Yes

Reviewer #3: (No Response)

6. Review Comments to the Author

Reviewer #1: Thanks once again for improving the quality of your work. I have no further comments to add on this work.

Reviewer #3: (No Response)

7. PLOS authors have the option to publish the peer review history of their article (what does this mean?). If published, this will include your full peer review and any attached files.

Reviewer #1: **Yes: **Sahabi Kabir Sulaiman

Reviewer #3: No

---

## [Editor Report · Acceptance letter]

26 Sep 2024

PONE-D-24-09721R1 

PLOS ONE

Dear Dr. Mwiinde, 

I'm pleased to inform you that your manuscript has been deemed suitable for publication in PLOS ONE. Congratulations! Your manuscript is now being handed over to our production team.

Kind regards, 

on behalf of

Dr. Richard Makurumidze 

Academic Editor

PLOS ONE